# McKittrick–Wheelock Syndrome, a Rare Cause of Nonresponsive Persistent Dyselectrolytemia

**DOI:** 10.3390/diagnostics15192459

**Published:** 2025-09-26

**Authors:** Daniela Cana Ruiu, Mihaela Cheie, Mirela Marinela Florescu, Andreea Doriana Stanculescu, Carmen Popescu, Daniela-Teodora Maria, Sebastian Constantin Toma, Naomi Fota, Daniela Calina, Bogdan Silviu Ungureanu

**Affiliations:** 1Department of Nephrology, University of Medicine and Pharmacy of Craiova, 200349 Craiova, Romania; danielacana.dc@gmail.com (D.C.R.); daniela.maria@umfcv.ro (D.-T.M.); naomi.fota@umfcv.ro (N.F.); 22nd Surgical Clinic, Craiova’s Emergency Hospital, 200349 Craiova, Romania; mihaela.cheie@umfcv.ro; 3Department of Pathology, University of Medicine and Pharmacy of Craiova, 200349 Craiova, Romania; mirela.florescu@umfcv.ro (M.M.F.); carmen.popescu@umfcv.ro (C.P.); 4Department of Anesthesia and Intensive Care, University of Medicine and Pharmacy of Craiova, 200349 Craiova, Romania; 5General Surgery Department, Dr. Constantin Andreoiu, Ploiesti County Emergency Hospital, 100097 Ploiesti, Romania; sebastian.toma@yahoo.com; 6Department of Clinical Pharmacy, University of Medicine and Pharmacy of Craiova, 200349 Craiova, Romania; daniela.calina@umfcv.ro; 7Department of Gastroenterology, University of Medicine and Pharmacy of Craiova, 200349 Craiova, Romania; bogdan.ungureanu@umfcv.ro

**Keywords:** hypokalemia, hyponatremia, villous adenoma

## Abstract

**Case Presentation: **A 67-year-old man presented with transient loss of consciousness and dizziness after weeks of vomiting, weakness, and recurrent syncopal episodes. Initial laboratory findings showed hyponatremia (Na 125 mEq/L), severe hypokalemia (K 2.3 mEq/L), hypochloremia (Cl 77 mEq/L), metabolic alkalemia (pH 7.5; HCO_3_^−^ 34 mEq/L), low serum osmolality (263 mOsm/L) with inappropriately concentrated urine (332 mOsm/kg), and prerenal azotemia (creatinine 3.4 mg/dL; eGFR 19 mL/min/1.73 m^2^; blood urea 209 mg/dL). Contrast-enhanced CT, along with colonoscopy, identified a large mucus-secreting villous adenoma in the upper rectum. After fluid and electrolyte replacement, the patient underwent surgical resection with complete remission of symptoms and correction of electrolyte abnormalities on follow-up. **Conclusion: **Rectal villous adenomas should be considered in older adults with unexplained hypovolemia, hypokalemic hyponatremia, and metabolic alkalemia. Early recognition and definitive resection are curative and prevent kidney injury.

## 1. Introduction

Electrolyte imbalances such as hyponatremia and hypokalemia are frequent clinical problems, often secondary to renal, endocrine, or iatrogenic causes. However, when these disturbances are profound, persistent, and resistant to standard corrective measures, they can signal an underlying and less obvious pathology requiring a broader diagnostic approach.

McKittrick–Wheelock syndrome (MKWS), also called electrolyte depletion syndrome [1], was first described in 1954 as benign villous adenomas, mostly located in the rectum or sigmoid colon, which secrete high quantities of electrolyte-rich mucin, and cause electrolyte’ loss, leading to hypovolemia, acute kidney injury (AKI), severe hypokalemia, and hyponatremia [2]. A systematic review study revealed a long duration of symptoms and severe electrolyte imbalances for patients with MKWS [3,4]. Because of symptom persistence and the first presumptions of other conditions, the diagnosis of MKWS is frequently delayed [4], especially if an infection such as Clostridioides difficile coexists [2]. Since these patients are at high risk of repeated admissions and recurrent AKI and electrolyte abnormalities [1], the only possibility remains endoscopic or surgical resection.

Due to its rarity and variable clinical expression, MKWS is often underdiagnosed or misattributed to more common conditions. Timely recognition is essential, as conservative management is rarely sufficient, and definitive treatment requires surgical resection of the secretory tumor.

This case report describes a diagnostically challenging case of MKWS from a nephrologist’s perspective, emphasizing the clinical reasoning, differential diagnosis, and therapeutic decision-making process that ultimately led to successful management.

## 2. Case Presentation

A 67-year-old male presented in the Emergency Department for craniofacial trauma after a transient loss of consciousness. Before admission, the patient’s symptoms were dizziness, repeated syncopes, vomiting unrelated to food, as well as physical asthenia. No relevant medical history or regimens were associated with the patient’s symptoms. Psychosocial history revealed no ethanol use and no history of smoking. The clinical examination revealed a moderately altered general condition, normal weight, hypovolemia, diffuse muscular hypotonia, without any osteoarticular anomalies, one watery stool every 2–3 days, and a diuresis of 2000 mL/24 h. Rectal exam was normal.

The biological status revealed the following anomalies: leukocytosis, significant inflammatory syndrome, metabolic alkalosis, hyponatremia, hypokalemia without urinary loss, serum hypoosmolality, significant nitrogen retention, severe hyperuricemia, moderate hyperaldosteronism, elevated values of prostate-specific antigen, and no changes in the digestive tumor markers (Table 1 and Appendix A). The electrocardiographic changes associated with hypokalemia were encountered with a long QT syndrome and ST segment depression.

Echocardiography did not reveal any pathological elements. Abdominal ultrasound revealed bilateral cortical kidneys with normal echogenicity, indicating no obstructive pathology. Due to persistent and symptomatic hyponatremia and hypokalemia, the patient was administered isotonic saline for plasma volume restoration, glucose solutions, 5.8% hypertonic saline, and 0.9% sodium chloride (NaCl), as well as 7.4% potassium chloride (KCl), anti-aldosterone antagonists, and oral sodium and potassium supplements. Correction of alkalosis with methionine was attempted to correct hypokalemia. Additional medications included hypouricemics, such as allopurinol, used to decrease high blood uric acid levels, considered together with hypovolemia, triggering factors of renal dysfunction. Renal function was initially restored. Despite the rebalancing therapies, the electrolyte changes relapsed during hospitalization. Low values of serum potassium and sodium persisted, and normalization was not achieved even after the administration of electrolyte solutions. However, the patient had no other relevant symptoms, no skin or urinary losses. Although the patient had only experienced watery stools every 2 to 3 days, we decided to perform a colonoscopy. The procedure revealed, at 10 cm from the anus, a large, circumferential, granular, nodular, mixed lateral spreading tumor that protruded into the lumen and covered a distance of approximately 12 cm (Figure 1). Mucin deposits were secreted by the giant polyp and were visible on the entire lesion. (Figure 1).

Further, we performed a computed tomography scan, which revealed a circumferential thickening of the rectal walls up to 16 mm and rectal dilatation, with no other significant abnormalities (Figure 2).

In all these clinical and paraclinical contexts, we concluded that the patient had MKWS. We initially considered the idea of endoscopic resection; however, due to the large extent of the lesion and its significant intrinsic development, we deemed it appropriate to recommend surgical resection of the villous adenoma. In the exploratory laparotomy, a moderate dilatation of the sigmoid and left colon was found macroscopically. At the level of the upper rectum, a soft, elastic tumor formation was detected, and a Dixon-type recto-sigmoid resection with mechanical end-to-end recto-sigmoid anastomosis and right iliac lateral ileostomy and appendectomy was performed. The surgical piece taken was represented by a colic loop of approximately 18 cm, sigmoid and upper and middle rectum, with a vegetative, occlusive tumor extending longitudinally over 12 cm, with the macroscopic appearance of a villous polyp (Figure 3).

The distal resection margin was not invaded, approximately 0.5 cm from the tumor border. Postoperatively, intestinal transit resumed at the level of the ileostomy, and digestive tolerance was good.

Pathology revealed a tubular adenoma (Figure 4) with high-moderate dysplasia, but also areas with aggravated cystic dysplasia and areas of malignancy with islands of moderately differentiated adenocarcinoma invading a third of the muscular tunic (Figure 5).

Ten days postoperatively, the intravenous administration of hydroelectrolytic rebalancing solutions was no longer necessary, with the ionogram, creatinine, and urea levels being in normal range, along with the acid–base balance. Two months after the initial surgical intervention, the ileostomy loop is excised, and an end-to-end entero-enteric anastomosis is performed, resulting in a favorable postoperative outcome.

## 3. Discussion

This case describes a patient presenting with simultaneous hypotonic hyponatremia, severe hypokalemia, and prerenal azotemia with syncope. This is an electrolyte pattern that mandates rapid assessment of tonicity and effective arterial blood volume. Our patient had a low serum osmolality with inappropriately concentrated urine, clinical hypovolemia, and secondary activation of the renin–angiotensin–aldosterone system (RAAS) [1,2,3]. Arterial blood gas analysis showed alkalemia; in this context, the alkalosis is best explained by chloride-responsive contraction alkalosis, superimposed on recent vomiting and ongoing extrarenal electrolyte losses.

We excluded common renal and endocrine causes of concurrent hypo-Na^+^/hypo-K^+^: no diuretics or corticosteroids, no thyroid/adrenal failure, normal magnesium (ruling out Mg^2+^-mediated refractory hypokalemia), and no evidence for tubular disorders. Pseudohypokalemia was dismissed when leukocytosis resolved without potassium correction. SIAD was unlikely given hypovolemia and a normal copeptin (a vasopressin surrogate) despite hypotonic hyponatremia [4,5,6,7]. The remaining unifying diagnosis had to be extrarenal losses, and the persistence of dyselectrolytemia after vomiting subsided pointed away from emesis as the sole driver [8,9].

Colonoscopy then provided the missing piece: a large villous adenoma of the upper rectum, accompanied by copious mucus. This fits McKittrick–Wheelock syndrome (MKWS)—the triad of a giant distal colorectal villous tumor, renal dysfunction from volume depletion, and profound electrolyte loss [10]. Secretory villous adenomas can generate up to ~4 L/day of mucoid effluent with Na^+^ concentrations of ~120 mmol/L and K^+^ concentrations of ~4.4 mmol/L, sufficient to produce the observed hypo-Na^+^/hypo-K^+^ and prerenal azotemia, despite limited overt stool frequency when discharge is retained distally [11,12]. Our patient’s mucus-predominant stools every 2–3 days illustrate how MKWS may masquerade as “not diarrheal” from the history yet produce continuous electrolyte losses biochemically. A 2018 review collated ~257 cases, underlining the syndrome’s rarity but also its consistent renal presentation; malignant transformation risk is 15–25% overall and ~40% when the lesion exceeds 4 cm, making definitive resection mandatory [12,13,14].

Published reports on McKittrick–Wheelock syndrome consistently describe a constellation of findings that mirror those observed here: disproportionate hypokalemia in the setting of hypotonic hyponatremia, urine osmolality that remains inappropriately elevated despite low serum osmolality (reflecting non-osmotic antidiuretic hormone release in hypovolemia), and rapid biochemical normalization following excision of the villous lesion. Acid–base disturbances vary across cases—metabolic alkalosis when vomiting and chloride depletion predominate versus metabolic acidosis when bicarbonate-rich colonic losses dominate—therefore, arterial blood gas results should be interpreted in conjunction with chloride balance and renin–angiotensin–aldosterone system (RAAS) activity rather than used in isolation to exclude the syndrome [11,12,13,14].

Management recommendations derived from the literature emphasize restoration of effective arterial volume with isotonic saline; aggressive potassium replacement with magnesium supplementation when indicated; targeted chloride repletion to correct chloride-responsive alkalosis; conservative limits on the rate of sodium correction to mitigate osmotic demyelination risk; avoidance of diuretics during resuscitation; and, critically, definitive removal of the secretory source [11,12,13,14]. Although tumor-related alterations in Na^+^/K-ATPase activity have been proposed, these remain speculative [15,16].

Patients with MKWS often present critically ill with hyponatremia, hypokalemia, and/or renal dysfunction [3], as seen in our case. The secretory activity is attributed to the overexpression of prostaglandin E_2_, which is particularly elevated in secretory adenomas and observed in fewer than 3% of large villous lesions [17]. COX inhibitors may temporarily reduce volume loss but should be used cautiously due to the risk of Clostridium difficile infection and overlapping diarrheal etiologies [2,18]. While MKWS lesions are often low- or high-grade dysplastic adenomas [19,20], adenocarcinoma is found in up to 18% of resected cases [21]. Definitive treatment involves surgical resection of the affected colorectal segment [22]. Options include conventional or laparoscopic surgery, transanal endoscopic surgery, and endoscopic submucosal dissection (ESD). ESD offers faster recovery and lower invasiveness but is preferred for smaller lesions due to the risk of post-procedural stenosis requiring dilation [23,24]. Transanal endoscopic approaches have also shown favorable outcomes [25].

If left untreated, secretory villous adenomas are associated with 100% mortality [3,26]. Among reported cases, mortality was 61.5% without surgery versus 5.2% with surgical management [3]. Our patient achieved full clinical and biochemical recovery following surgical excision. Given the malignant potential of these lesions, postoperative endoscopic surveillance remains essential [27]. This case underscores the importance of considering MKWS in patients with persistent, unexplained electrolyte disturbances—even in the absence of typical secretory diarrhea. From a nephrologist’s standpoint, recognizing paraneoplastic secretory syndromes can facilitate timely diagnosis and definitive treatment, ultimately improving patient outcomes.

The importance of an algorithm for diagnosing hyponatremia and hypokalemia lies in its ability to bring systematic structure, efficiency, and clinical accuracy to managing a potentially complex and life-threatening electrolyte disorder. An algorithm transforms a complex, multifactorial electrolyte disorder into a step-by-step diagnostic approach—improving accuracy, efficiency, education, and patient outcomes (Figure 6).

## 4. Conclusions

MKWS is a clinical entity that warrants special attention due to its nonspecific symptomatology, which overlaps with many other clinical situations, thus delaying diagnosis and, consequently, the implementation of appropriate therapeutic measures. The hydroelectrolytic abnormalities associated with this syndrome increase the risk of major events and the mortality of these patients in the absence of appropriate therapy. Thus, the hydroelectrolytic rebalancing therapy, which is necessarily followed by endoscopic and/or surgical resection, must be prompt and adequate.

The multidisciplinary approach of this syndrome, by gastroenterologists, nephrologists, surgeons, and sometimes other clinicians, is vital for the prognosis of these patients.

## Figures and Tables

**Figure 1 diagnostics-15-02459-f001:**
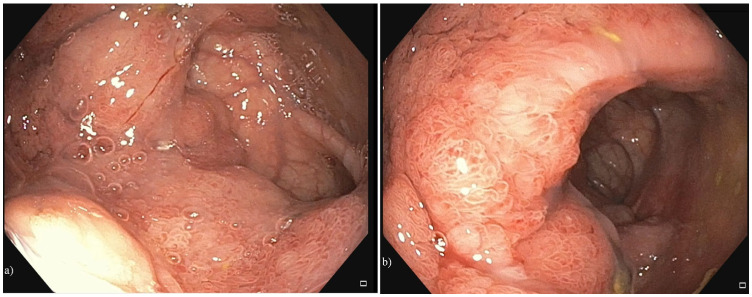
(**a**,**b**) The lateral spreading tumor with a granular aspect, classified as Paris IIa+Is, Kudo IV.

**Figure 2 diagnostics-15-02459-f002:**
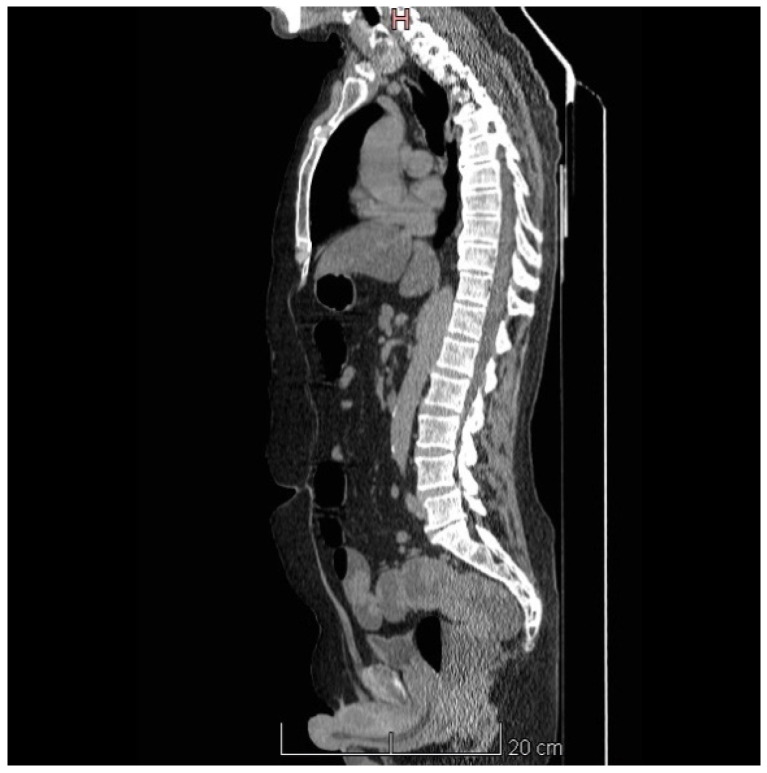
Sagittal CT of the abdomen-pelvis shows a long segment of circumferential thickening of the upper rectal wall.

**Figure 3 diagnostics-15-02459-f003:**
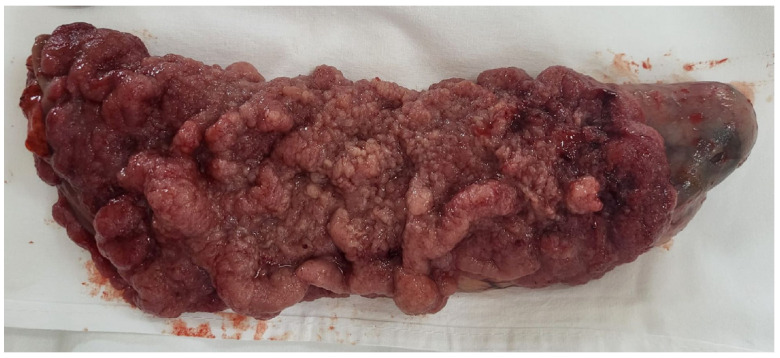
Macroscopic image of the lesion, suggesting a carpet-like circumferential villous mucosal tumor. The surface is tan-pink to red with coalescent nodules and focal superficial hemorrhage, consistent with a large villous adenoma.

**Figure 4 diagnostics-15-02459-f004:**
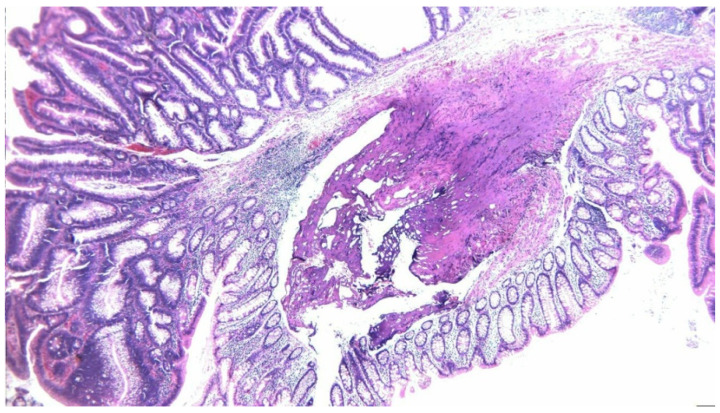
Tubulo-villous adenoma with intact resection margins, HE, 10×.

**Figure 5 diagnostics-15-02459-f005:**
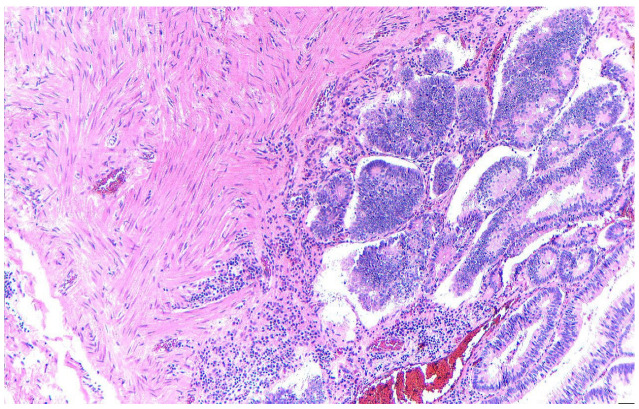
Tubular adenoma with invasive adenocarcinoma area in the muscularis (HE, 10×).

**Figure 6 diagnostics-15-02459-f006:**
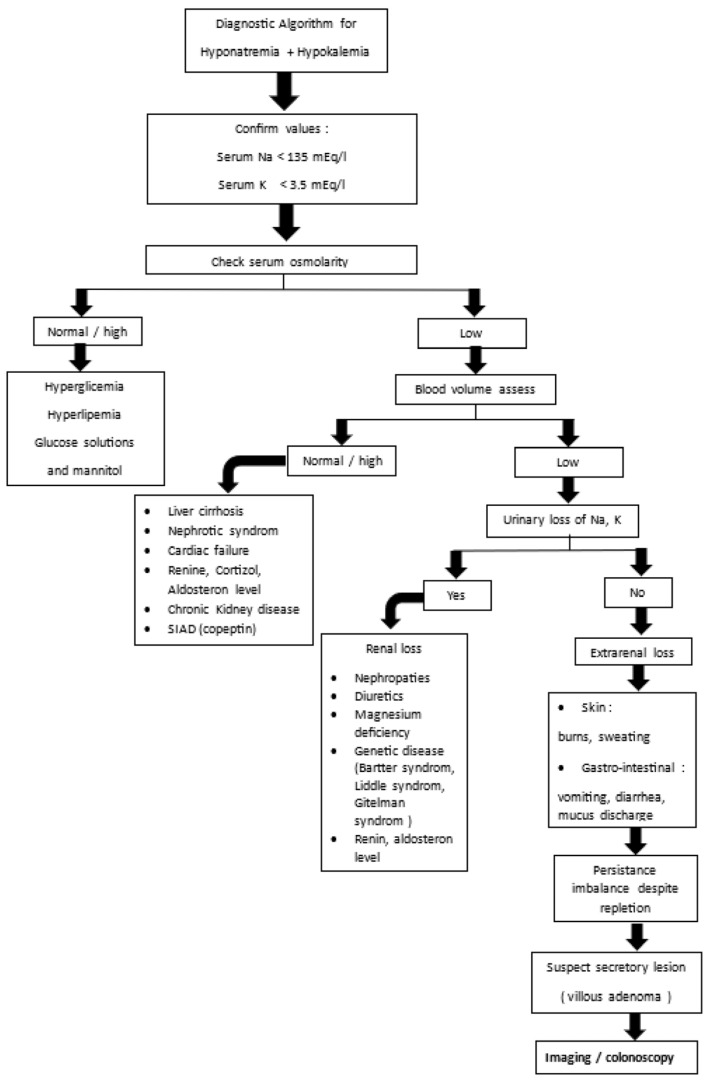
Diagnostic work-up for Hyponatremia + Hypokalemia.

**Table 1 diagnostics-15-02459-t001:** Essential biological test used for diagnostic reasoning.

Parameter	Value	Reference Range
Sodium (mEq/L)	125	135–145
Potassium (mEq/L)	2.3	3.5–5.1
Chloride (mEq/L)	77	98–107
pH (arterial)	7.50	7.35–7.45
Bicarbonate, HCO_3_^−^ (mEq/L)	34	22–29
PaCO_2_ (mmHg)	47	35–45
Creatinine (mg/dL)	3.4	<1.2
eGFR (mL/min/1.73 m^2^)	19	>90
Urea (mg/dL)	209	13–43
Serum osmolality (mOsm/kg)	263	275–295
Urine osmolality (mOsm/kg)	332	300–900
Magnesium (mg/dL)	2.37	1.7–2.4

## Data Availability

The data of study is available from corresponding authors upon request.

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
