# Peer review of "McKittrick–Wheelock Syndrome, a Rare Cause of Nonresponsive Persistent Dyselectrolytemia"

_diagnostics, 2025, doi:10.3390/diagnostics15192459_

Round 1
Reviewer 1 Report (New Reviewer)
Comments and Suggestions for Authors
Thanks to the Authors for this interesting case report. I suggest a few minor revisions and clarifications in order to get ghe text more precise:
- case presentation. In the first paragraph, please specify what do you mean for "diuresis 2000ml" (in 24 hours? When?) and please better define the characteristics of the "significant inflammatory syndrome" regarding laboratory test (second paragraph)
- Table 1, acide-base and electrolyte balance. Please specify the pH
- Page 5. I suppose it lacks the word "endoscopic" when talking about "the first idea of resection"
Author Response
We sincerely appreciate the positive feedbacks from the reviewers. We have made every effort to thoroughly address the Reviewers’ helpful comments and suggestions. Please see our responses below
Thanks to the Authors for this interesting case report. I suggest a few minor revisions and clarifications in order to get the text more precise:
- case presentation. In the first paragraph, please specify what do you mean for "diuresis 2000ml" (in 24 hours? When?) and please better define the characteristics of the "significant inflammatory syndrome" regarding laboratory test (second paragraph)
Thank you for the remark, we mentioned it was on 24hours.
- Table 1, acide-base and electrolyte balance. Please specify the pH
We inserted the value from the Astrup Work-up
- Page 5. I suppose it lacks the word "endoscopic" when talking about "the first idea of resection"
Thank you for your observation. We modified accordingly

Reviewer 2 Report (New Reviewer)
Comments and Suggestions for Authors
McKittrick-Wheelock Syndrome, a Rare Cause of Nonrespon-sive Persistent Dyselectrolytemia
General Assessment
The manuscript presents a case of McKittrick–Wheelock syndrome, a rare but interesting and relevant condition. However, the overall structure and quality of the paper require substantial revision before it can be considered for publication.
MAJOR CONCERNS
Length and redundancy: The case description is disproportionately long and includes details that do not add value to the clinical understanding of the case. The discussion is overly verbose and at times repetitive, which dilutes the key clinical messages.
Superfluous data: Numerous laboratory parameters are presented in exhaustive detail (e.g., multiple urine and serum markers, extended biochemical profiles) without adequate interpretation or clear relevance to the diagnosis or management of McKittrick–Wheelock syndrome.
. Tables I and II are particularly overloaded and should be reduced to the essential data supporting the diagnostic reasoning (sodium, potassium, bicarbonate, renal function). PS: a) please provide information on blood pH and pCO2; b) put together the two tables in a simplified table.
Poor English Language Quality: The manuscript is written in suboptimal English, with frequent grammatical errors, awkward phrasing, and inconsistent terminology (e.g., 'hydroelectrolyte imbalance' instead of 'electrolyte imbalance'). The text would greatly benefit from thorough language editing.
SPECIFIC COMMENTS
- Abstract: Too detailed for a case report abstract. It should be condensed to include the main clinical presentation, diagnosis, treatment, and outcome.
PS: Provide quantitative information such as sodium level, potassium level, acid-base balance, and kidney function.
- Figures: Some images (e.g., endoscopic and macroscopic views) are poorly described and would benefit from concise legends highlighting their clinical significance. Figure 6 is rather complex and essentially of no real value.
- Discussion: Overloaded with textbook-style explanations of hyponatremia and hypokalemia, which are not directly linked to the presented case. This section should be shortened and focused on how the presented case aligns with or differs from previously published reports on McKittrick–Wheelock syndrome.
- Furthermore: Have the manuscript edited by a professional English language editor to ensure clarity and readability.
Comments on the Quality of English Language
See comments for authors
Author Response
Thank you for the kind advice. We have tried our best to improve the manuscript and made some changes. We hope that the correction will be well-recieved. Thank you and best regards.
McKittrick-Wheelock Syndrome, a Rare Cause of Nonresponsive Persistent Dyselectrolytemia
General Assessment
The manuscript presents a case of McKittrick–Wheelock syndrome, a rare but interesting and relevant condition. However, the overall structure and quality of the paper require substantial revision before it can be considered for publication.
MAJOR CONCERNS
Length and redundancy: The case description is disproportionately long and includes details that do not add value to the clinical understanding of the case. The discussion is overly verbose and at times repetitive, which dilutes the key clinical messages.
Superfluous data: Numerous laboratory parameters are presented in exhaustive detail (e.g., multiple urine and serum markers, extended biochemical profiles) without adequate interpretation or clear relevance to the diagnosis or management of McKittrick–Wheelock syndrome.
We moved the tables to a supplementary material file
. Tables I and II are particularly overloaded and should be reduced to the essential data supporting the diagnostic reasoning (sodium, potassium, bicarbonate, renal function). PS: a) please provide information on blood pH and pCO2; b) put together the two tables in a simplified table.
We accepted the suggestion to add only parameters that materially aid the McKittrick–Wheelock diagnosis. The merged table now includes chloride, urea, serum/urine osmolality, and magnesium in addition to sodium, potassium, bicarbonate, pH/PaCO₂, and renal function. These additions clarify volume status, tonicity, and prerenal azotemia without re-overloading the table.”
Poor English Language Quality: The manuscript is written in suboptimal English, with frequent grammatical errors, awkward phrasing, and inconsistent terminology (e.g., 'hydroelectrolyte imbalance' instead of 'electrolyte imbalance'). The text would greatly benefit from thorough language editing.
We did our best to improve the manuscript and hoped to have corrected the grammatical errors.
SPECIFIC COMMENTS
- Abstract: Too detailed for a case report abstract. It should be condensed to include the main clinical presentation, diagnosis, treatment, and outcome.
PS: Provide quantitative information such as sodium level, potassium level, acid-base balance, and kidney function.
Response: We rewrote the abstract to ~160 words, focusing on presentation, key diagnostics (CT/colonoscopy), definitive treatment (surgical resection), and outcome (resolution of symptoms and electrolyte correction).
Response: We added specific values to the abstract: Na 125 mEq/L, K 2.3 mEq/L, Cl 77 mEq/L, pH 7.5, HCO₃⁻ 34 mEq/L, serum osmolality 263 mOsm/L, urine osmolality 332 mOsm/kg, creatinine 3.4 mg/dL (eGFR 19 mL/min/1.73 m²), and blood urea 209 mg/dL, all drawn from the laboratory tables in the manuscript.
A 67-year-old man presented with transient loss of consciousness and dizziness after weeks of vomiting, weakness, and recurrent syncopal episodes. Initial laboratory findings showed hyponatremia (Na 125 mEq/L), severe hypokalemia (K 2.3 mEq/L), hypochloremia (Cl 77 mEq/L), metabolic alkalemia (pH 7.5; HCO₃⁻ 34 mEq/L), low serum osmolality (263 mOsm/L) with inappropriately concentrated urine (332 mOsm/kg), and prerenal azotemia (creatinine 3.4 mg/dL; eGFR 19 mL/min/1.73 m²; blood urea 209 mg/dL). Contrast-enhanced CT followed by colonoscopy identified a large mucus-secreting villous adenoma in the upper rectum. After fluid and electrolyte replacement, the patient underwent surgical resection with complete remission of symptoms and correction of electrolyte abnormalities on follow-up.
Conclusion: Rectal villous adenomas should be considered in older adults with unexplained hypovolemia, hypokalemic hyponatremia, and metabolic alkalemia. Early recognition and definitive resection are curative and prevent kidney injury.
- Figures: Some images (e.g., endoscopic and macroscopic views) are poorly described and would benefit from concise legends highlighting their clinical significance. Figure 6 is rather complex and essentially of no real value.
We modified the image description as recommended. We agree with your opinion and provided an improved description
However, we did not remove Figure 6, as we considered that it might be of assistance to young physicians who may encounter similar cases. If you still consider that we should remove it, we will include in the Supplementary Material File.
- Discussion: Overloaded with textbook-style explanations of hyponatremia and hypokalemia, which are not directly linked to the presented case. This section should be shortened and focused on how the presented case aligns with or differs from previously published reports on McKittrick–Wheelock syndrome.
We downsized the discussion as recommended and provided a more thorough approach.
- Furthermore: Have the manuscript edited by a professional English language editor to ensure clarity and readability.
- We did our best to improve the manuscript and hoped to have corrected the grammatical errors.
We appreciate your advice and approach, and hope to have fulfilled the requirements.
Round 2
Reviewer 2 Report (New Reviewer)
Comments and Suggestions for Authors
The revised manuscript has certainly improved, and I commend the authors.
I recommend three further changes:
a) The case presentation section could be made at least 15% more concise.
b) In the final part of the case presentation section, it's important to note that the sodium, potassium, creatinine, and urea levels, in addition to the acid-base balance, were all normal.
c) In the table, the pH value should precede that of pCO2 and the bicarbonate level.
Author Response
Thank you for taking your time to check the manuscript changes we made.
We took into account your observation, and tried to simplify the text and make it more objective by eliminating sentences such as the one in italic:
Echocardiography did not reveal pathological elements (no cavity or kinetic abnormalities, preservation of the ejection fraction of the left ventricle).
which explains the elevated aldosterone and ongoing renal potassium wasting typical of volume depletion rather than primary mineralocorticoid excess (absence of hypertension or edema)
and the persistence of dyselectrolytemia after vomiting subsided pointed away from emesis as the sole driver
etc.
We also modified the table and added the idea of normalized biological status after surgery.
We appreciate your recommendations and are grateful for taking the time to help us improve our article.
This manuscript is a resubmission of an earlier submission. The following is a list of the peer review reports and author responses from that submission.
Round 1
Reviewer 1 Report
Comments and Suggestions for Authors
Despite the work of the authors this paper does not bring anything new in the field. It is a classical MKWS.
For a Case Report there are to many authors.
Author Response
Dear reviewer,
While we acknowledge the case may seem straightforward, we believe it offers important educational insights, especially for clinicians who may encounter similar presentations. Its value lies in reinforcing diagnostic vigilance and appropriate management in routine clinical settings.
That is why we focused more on the discussion section and provided more insights from a nephrologist's point of view.
All the authors included in the manuscript have interacted in one way or another with the case, and it was only fair that they should have been included in the manuscript. However, one of the authors confirmed that he may withdraw his authorship with no problem whatsoever.
Our case underscores a common clinical pitfall that may be overlooked. By sharing this case, we aim to raise awareness and contribute to improved patient outcomes in everyday practice.
Reviewer 2 Report
Comments and Suggestions for Authors
This report presents a case of combined electrolyte abnormalities and renal dysfunction secondary to intestinal losses from a bowel syndrome corrected by surgical excision of the tumor. The Discussion section presents a detailed evaluation of the case and the pathophysiology of electrolyte abnormalities. I have a few suggestions that can assist in the presentation of this report.
-Line 63: define the features of "dehydration syndrome" found in this patient. Please note that several reports have stressed the need to differentiate between "dehydration" and "hypovolemia". I suggest that "hypovolemia", that is low extracellular volume caused by loss of sodium salts primarily and water accompanying the salt losses, may be preferable than "dehydration" in this case given the hypotonic hyponatremia which indicates water excess, not deficit, in relation to body fluid effective solutes.
-Table 2. (a) Was the entity measured serum bicarbonate or serum total carbon dioxide (TCO2)? Modern autoanalyzers measure TCO2. (b) Please change "Chlorum" to "chloride". (c) Please change "osmolarity" to "osmolality" for both the blood and urine. The term "osmolality" should be used for the quantity measured by a colligative method, such as the freezing point, and reported in mOsm/kg H2O. The term "osmolarity" is used for the quantity calculated, not measured, as the sum of concentrations in serum of its key solutes. Serum osmolarity, expressed in mmol/L or mOsm/L, is calculated by formulas, most commonly by the formula 2xsodium + Glucose + urea, all expressed in mmol/L. The difference between serum osmolality and osmolarity, the osmol gap, is used to diagnose pseudohyponatremia or the presence in the serum of solutes other than urea and glucose. If serum urea and glucose were measured, their measurements should be added to the Table. The diagnosis of hypotonic hyponatremia would be strengthened.
-Line 97. The high value of serum bicarbonate (or TCO2) could be caused by metabolic alkalosis or respiratory acidosis. If blood gas measurement was made, its results should be shown. If not, a comment on the patient's respiratory system should be added. Absence of significant respiratory disease would strengthen the diagnosis of metabolic alkalosis.
-Line 231. "...copeptin was normal". Please clarify. Was the copeptin level of the patient presented normal? If this is the case, the copeptin level or how the decision that this level was normal was taken should be presented.
-Please add a sentence in the Discussion stating that Figure 6 presents the scheme for the diagnostic approach to hyponatremia and hypokalemia.
Comments on the Quality of English Language
The English of some passages can be improved. I will present one example: The English of the sentence between lines 130 and 132 would be improved as follows: "In the exploratory laparotomy, a moderate dilatation of the sigmoid and left colon was found macroscopically." The presentation of this report can be improved if the authors obtain the assistance of a person fluent in medical English writing.
Author Response
Dear reviewer,
Thank you for taking the time to review our manuscript and for your valuable feedback. We appreciate your thoughtful comments and have addressed each point carefully below.
Point-by-point reply to the Reviewers’comments
-Line 63: define the features of "dehydration syndrome" found in this patient. Please note that several reports have stressed the need to differentiate between "dehydration" and "hypovolemia". I suggest that "hypovolemia", that is low extracellular volume caused by loss of sodium salts primarily and water accompanying the salt losses, may be preferable than "dehydration" in this case given the hypotonic hyponatremia which indicates water excess, not deficit, in relation to body fluid effective solutes.
Our patient has a loss of extracellular fluid (plasma volume), including both water and electrolytes ( sodium and potassium), so "hypovolemia", that is, low extracellular volume caused by loss of sodium salts primarily and water accompanying the salt losses, is preferable to "dehydration". Thank you for your comment. We agree and have modified accordingly
-Table 2. (a) Was the entity measured serum bicarbonate or serum total carbon dioxide (TCO2)? Modern autoanalyzers measure TCO2.
The entity measured was serum bicarbonate. The total carbon dioxide was not measured. In our laboratory, it is not possible to determine TCO2, except in intensive care units; the patient did not require intensive care monitoring.
Thank you for your suggestion!
(b) Please change "Chlorum" to "chloride".
We have changed "Chlorum" to "chloride". The information refers to the chloride ion (Cl⁻), a key electrolyte involved. Thank you for pointing this out.
(c) Please change "osmolarity" to "osmolality" for both the blood and urine. The term "osmolality" should be used for the quantity measured by a colligative method, such as the freezing point, and reported in mOsm/kg H2O. The term "osmolarity" is used for the quantity calculated, not measured, as the sum of concentrations in serum of its key solutes. Serum osmolarity, expressed in mmol/L or mOsm/L, is calculated by formulas, most commonly by the formula 2xsodium + Glucose + urea, all expressed in mmol/L. The difference between serum osmolality and osmolarity, the osmol gap, is used to diagnose pseudohyponatremia or the presence in the serum of solutes other than urea and glucose. If serum urea and glucose were measured, their measurements should be added to the Table. The diagnosis of hypotonic hyponatremia would be strengthened.
Thank you for your comment regarding the distinction between osmolarity and osmolality. We have changed "osmolarity" to "osmolality for urine. It was reported in mOsm/kg H2O. But for blood it was performed serum osmolarity, expressed as you mentioned in mmol/L or mOsm/L, calculated by formula 2xsodium + Glucose + urea, all expressed in mmol/L.
However, I have changed the unit of measurement from mOsm/Kg to mOsm/L We have also added the urea and glucose values in mmol/L to the table.
-Line 97. The high value of serum bicarbonate (or TCO2) could be caused by metabolic alkalosis or respiratory acidosis. If blood gas measurement was made, its results should be shown. If not, a comment on the patient's respiratory system should be added. Absence of significant respiratory disease would strengthen the diagnosis of metabolic alkalosis.
Also blood gas measurement was not made in our department, but we can confirm absence of significant respiratory disease. We consider it was a metabolic alkalosis and not a respiratory one.
-Line 231. "...copeptin was normal". Please clarify. Was the copeptin level of the patient presented normal? If this is the case, the copeptin level or how the decision that this level was normal was taken should be presented.
We have added the determined copeptin value to the table. Thank you!
-Please add a sentence in the Discussion stating that Figure 6 presents the scheme for the diagnostic approach to hyponatremia and hypokalemia.
We have added a comment in the Discussion stating that Figure 6 presents the scheme for the diagnostic approach to hyponatremia and hypokalemia. Every figure need to be cited in the body of the manuscript. We have modified to emphasize this point.
We appreciate your guidance in refining our manuscript and believe that these revisions have greatly improved the presentation and readability of the case report. Thank you once again for your valuable feedback.
Comments on the Quality of English Language
The English of some passages can be improved. I will present one example: The English of the sentence between lines 130 and 132 would be improved as follows: "In the exploratory laparotomy, a moderate dilatation of the sigmoid and left colon was found macroscopically." The presentation of this report can be improved if the authors obtain the assistance of a person fluent in medical English writing.
We have made the suggested edits to enhance clarity and coherence and have thoroughly corrected spelling, grammar, and syntax errors with the assistance of a native English speaker. We appreciate your guidance in improving our manuscript.
We appreciated the opportunity to refine our work under your expert guidance and looked forward to your further assessment.
We have incorporated all requested changes and revisions directly into the text, highlighting these in blue for clear visibility.
Round 2
Reviewer 1 Report
Comments and Suggestions for Authors
-